



# Global coarse grained mesoscale eddy statistics based on integrated kinetic energy and enstrophy correlations

Imre M. Jánosi[1,2], Holger Kantz[1], Jason A. C. Gallas[1,3], and Miklós Vincze[4,5]

[1]Max Planck Institute for the Physics of Complex Systems, Nöthnitzer Str. 38, 01187 Dresden, Germany
[2]University of Public Service, Faculty of Water Sciences, Department of Water and Environmental Policy, Ludovika tér 2, H-1083 Budapest, Hungary
[3]Instituto de Altos Estudos da Paraíba, Rua Silvino Lopes 419-2502, 58039-190 João Pessoa, Brazil
[4]von Kármán Laboratory for Environmental Flows, Eötvös Loránd University, Pázmány Péter s. 1/A, H-1117 Budapest, Hungary
[5]MTA-ELTE Theoretical Physics Research Group, Pázmány Péter s. 1/A, H-1117 Budapest, Hungary

**Correspondence:** Imre M. Jánosi (janosi.imre.miklos@uni-nke.hu)

**Abstract.** Recently, Jánosi et al. introduced the concept of a "super vortex proxy" based on an observation of strong correlations between integrated kinetic energy and integrated enstrophy over a large enough surface area. When mesoscale vortices are assumed to exhibit a Gaussian shape, the two spatial integrals have particularly simple functional forms, and a ratio of them defines an effective radius of a "proxy vortex". In the original work, the idea was tested over a restricted area in the Californian

Current System. Here we extend the analysis to global scale by means of 25 years of AVISO altimetry data covering the (ice free) global ocean. The results are compared with a global vortex data base containing over 64 million of mesoscale eddies. We demonstrate that the proxy vortex representation of surface flow fields also works globally and provides a quick and reliable way to obtain coarse grained vortex statistics. Estimated mean eddy sizes (effective radii) are extracted in very good agreement with the data from the vortex census. Recorded eddy amplitudes are directly used to infer the part of kinetic energy transported

by the mesoscale vortices. The ratio of total and eddy kinetic energies is somewhat higher than found in previous studies. The characteristic westward drift velocities are evaluated by a time lagged cross-correlation analysis of the kinetic energy fields. While zonal mean drift speeds are in good agreement with vortex trajectory evaluation in the latitude bands $30°S - 5°S$ and $5°N - 30°N$, discrepancies are exhibited mostly at higher latitudes on both hemispheres. A plausible reason of somewhat different drift velocities obtained by eddy tracking and cross-correlation analysis is the fact that the drift of mesoscale eddies is only one

component of the surface flow fields. Rossby wave activities, coherent currents, and other propagating features on the ocean surface apparently contribute to the zonal transport of kinetic energy.

## 1   Introduction

Mesoscale eddies (MEs) at spatial scales from approximately 50 up to 500 km are energetic patterns of ocean surface flow fields. The birth of MEs occurs often along shorelines triggered off by shear driven barotropic instabilities, or at the edges

of surface currents by density anomaly driven baroclinic instabilities (Willett et al., 2006; Chelton et al., 2007; Smith, 2007; Badin et al., 2009; Chelton et al., 2011; Faghmous et al., 2015; Marta and Isachsen, 2018; Brach et al., 2018; Pnyushkov et al.,





2018; Cetina-Heredia et al., 2019; van Sebille et al., 2020; Chérubin et al., 2021; Wichmann et al., 2021). The observation of mesoscale vortices on global ocean surfaces is usually based on satellite altimetry which determines local sea surface heights with respect to the geoid by return time analysis of reflected microwave pulses (Stammer and Cazenave, 2017). Local sea level

anomalies (SLA) are obtained by removing local long-term mean SSH values. 2D velocity fields are obtained by assuming geostrophic equilibrium where horizontal (hydrostatic) pressure gradient forces are compensated by the Coriolis effect. Several Eulerian methods have been developed to identify and locate mesoscale vortices from surface flow fields, most of them are based some form of finding close contours of SLA (Chelton et al., 2011; Mason et al., 2014; Li et al., 2016; Schütte et al., 2016; Pessini et al., 2018; Zhibing et al., 2022). Alternatives use the geometry of the velocity vectors (Nencioli et al., 2010; Ji et al.,

2018), contours of the Okubo-Weiss parameter (Chelton et al., 2007; Kurian et al., 2011; Ubelmann and Fu, 2011; Schütte et al., 2016; Pessini et al., 2018), or wavelet analysis (Rubio et al., 2009; Pnyushkov et al., 2018). Detailed comparisons show that all Eulerian methods have pros and cons, none of them is superior to another (Souza et al., 2011; Escudier et al., 2016). Methods based on identifying Lagrangian coherent structures obey a much better mathematical foundation (Haller, 2015; Beron-Vera et al., 2018; Haller et al., 2018; El Aouni, 2021; Ryzhov and Berloff, 2022). However, they are computationally

rather demanding and the (relatively low) spatial resolution of the input fields is a challenging part of them (Amores et al., 2018).

Kinetic energy (KE) is a quantitative characteristics of ocean flow fields (Wunsch and Ferrari, 2004; Stammer, 1997; Wunsch, 2009, 2013). KE is usually separated into the mean KE and the eddy KE (EKE) computed from the time-varying velocities, see Fig. 1. Recent data evaluation of satellite observations suggest that eddy-rich regions exhibit a significant increase in

mesoscale variability (thus EKE per unit volume), while the equatorial oceans show a decrease in EKE (Martínez-Moreno et al., 2019, 2021).

In this work we extend a previous analysis by Jánosi et al. (2019) to global scale based on the observation that integrated EKE and integrated enstrophy over a large enough area are strongly correlated in time. A coarse grained approach can reveal useful information such as mean eddy size or fraction of vortex energy in total EKE, similarly to a recent study by (Rai et al.,

2021) where oceanic eddy killing by wind was analysed. In the next Section we briefly summarize the essential points of the aforementioned methodology. Then we demonstrate the presence of strong correlations between integrated kinetic energy and enstrophy, and obtain effective "proxy vortex" radii globally. (In Jánosi et al. (2019), the term "super vortex" was coined to characterize the surface flow field by a single Gaussian vortex, here we rather use "proxy vortex" to avoid overstatement.) Note that we restrict our analysis to 2D surface flow fields, vertical vortex structures are not considered. We validate the procedure

by comparing the results with data on 64 millions of mesoscale vortices identified by the most common closed contour method Faghmous et al. (2015). In addition, we analyze the amplitude relationships between our method and the global vortex census. Finally, we provide a global survey of westward drift constructed by measuring temporal cross correlations of kinetic energies, discrepancies are discussed.

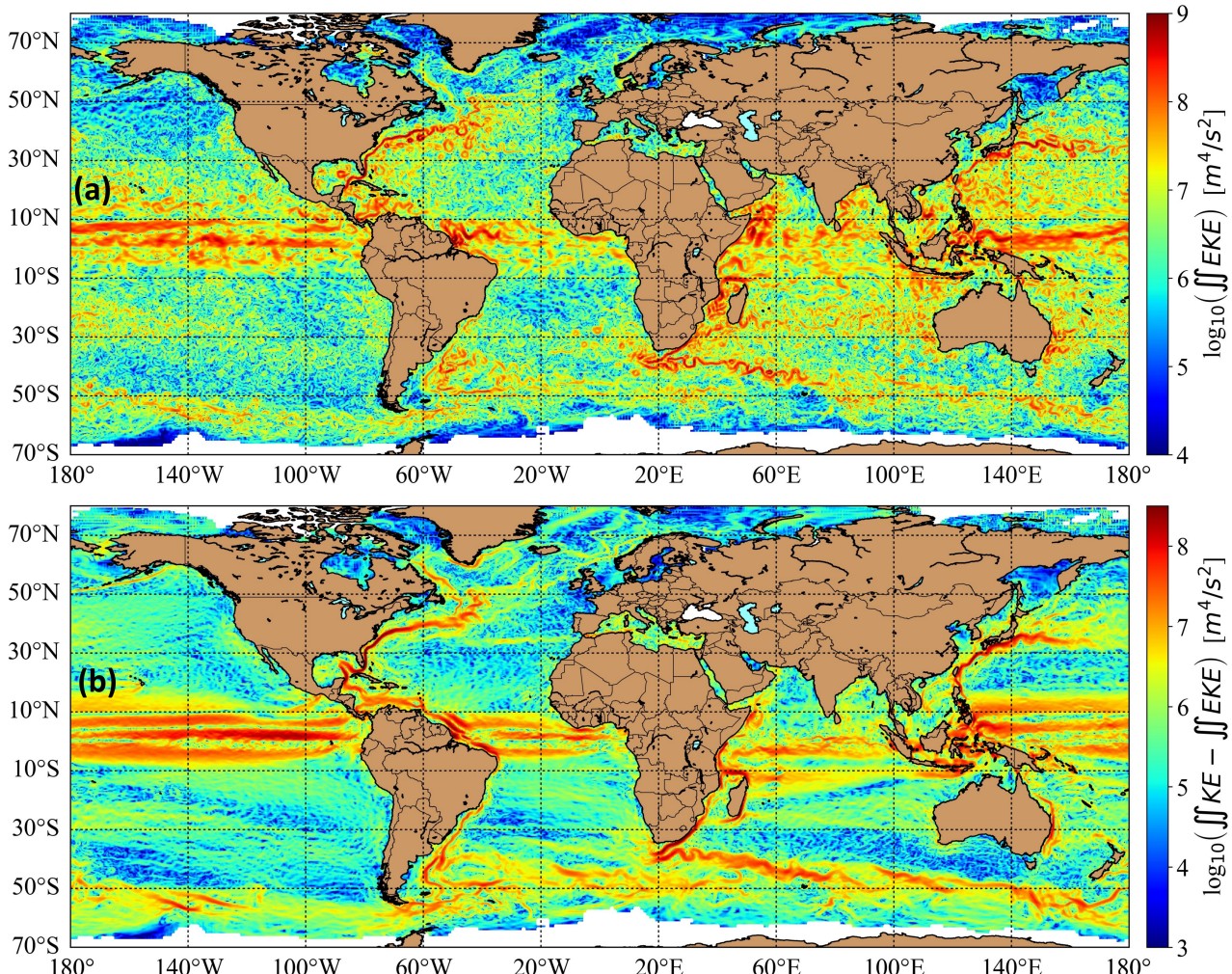

**Figure 1.** Snapshot of the global geostrophic flow field on a randomly chosen day (23 Sept 2018) from the AVISO data bank (Aviso, 1993-2018; Taburet et al., 2019). Color coding is in a logarithmic scale for a better visualization; note the different ranges. **(a)** Integrated eddy kinetic energy for each grid cell of $0.25° \times 0.25°$ from geostrophic velocity anomalies $[u'_g, v'_g]$, in units of $\mathrm{m^4 s^{-2}}$. **(b)** The difference between the integrated total kinetic energy from geostrophic velocities $[u_g, v_g]$ and the integrated eddy kinetic energy plotted in (a). Integration is performed by multiplying the squared velocity with the curvature corrected cell size measured in units of $\mathrm{m^2}$.

## 2 Data and methods

### 2.1 Data sources

Our primary data source is the AVISO data bank (Aviso, 1993-2018; Taburet et al., 2019). Besides sea level anomalies, geostrophic velocity components and their anomalies are also available. Absolute zonal and meridional velocity components




(`ugos` and `vgos` in AVISO, denoted here by $[u_g, v_g]$) were derived from sea surface height (SSH) data above geoid with the geostrophic balance relations. The absolute velocity components are related with the absolute dynamic topography, which is the sum of sea level anomaly and the mean absolute dynamic topography (MDT). MDT is one of the key quantities to characterize persistent ocean surface currents. Geostrophic velocity anomalies (`ugosa` and `vgosa`, $[u_g', v_g']$) were computed by removing twenty-year mean values (over the period of 1993 - 2012) for each grid cell (Aviso, 1993-2018; Taburet et al., 2019). In general, geostrophic velocity anomalies are related to mesoscale eddies representing deviations from the mean flow (Frenger et al., 2015; Ji et al., 2018). Fig. 1a illustrates the geographic distribution of eddy kinetic energy obtained from geostrophic velocity anomalies $[u_g', v_g']$, while Fig. 1b is determined from the difference between absolute velocities $[u_g, v_g]$ and anomalies $[u_g', v_g']$. Fig. 1a is supposed to characterize eddy activities, while Fig. 1b is related to the persistent energetic ocean currents. The overlap of red bands in Fig. 1a and Fig. 1b indicates intense vortex shedding along the major currents, however it cannot be separated from the effects of meandering and relocation of the main currents. The spatial resolution of AVISO fields is $0.25° \times 0.25°$ ($1440 \times 720$ grid cells), land areas are masked. The temporal resolution is 1 day in the period 1 Jan 1993 - 23 Oct 2018, 9397 days without missing dates. Total and eddy kinetic energies ($KE$ and $EKE$) are obtained trivially from $[u_g, v_g]$ and $[u_g', v_g']$, enstrophy (the squared vorticity) is determined from the curl of velocity field (by centered numerical derivatives). Since this operation does not work directly along the shorelines (at least one grid cell should be omitted), we matched the fields properly for an appropriate comparison.

For the validation, we exploited the vortex data bank assembled by Faghmous et al. (2015). They determined mesoscale eddies from the same AVISO data that we utilized, but covering a somewhat shorter temporal period (1 Jan 1993 - 2 May 2014, 7791 days). During this period they identified globally 32,687,988 cyclonic and 31,872,899 anticyclonic eddies. Several parameters are stored about each vortex (geographic location, size, major and minor axis lengths from a fit to an ellipsoid, major axis orientation, amplitude, area, etc.,) however we used a limited subset of such data, as explained in Section 3 (Results and discussions).

## 2.2 Gaussian mesoscale eddies

Several studies of the shape of ocean MEs revealed that they are close to Gaussian humps or troughs (Hopfinger and van Heijst, 1993; Chelton et al., 2011; Raj et al., 2016; Keppler et al., 2018; Martínez-Moreno et al., 2019). A detailed fitting procedure of about five million SLA profiles by Wang et al. (2015) revealed that around 50% of MEs are indeed well approximated by a Gaussian shape:

$$\eta(r) = \eta_0 \exp\left(-\frac{r^2}{2R^2}\right) , \tag{1}$$

where $\eta_0$ is the peak height, $r$ is the radial distance from the vortex center, and $R$ is a radius parameter. Note that $R$ belongs to the $1\sigma$ standard deviation of a Gaussian profile, which is not necessarily identified as the radius of an ME by closed contour methods. Besides the Gaussian eddies, another ~40% are Gaussian over a sloping background or merger of two nearby Gaussian eddies, and the rest have a quadratic core resembling Rankine vortices. Geostrophic equilibrium velocities in polar



coordinates have only nonzero tangential components $\mathbf{v}_g = [0, v_g(r), 0]$:

$$v_g(r) = \frac{g}{f}\frac{\partial \eta(r)}{\partial r} = -\frac{\eta_0 g r}{f R^2}\exp\left(-\frac{r^2}{2R^2}\right) \quad , \tag{2}$$

where $g$ is the gravitational acceleration, and $f = 2\Omega\sin(\varphi)$ is the local Coriolis parameter at latitude $\varphi$ with $\Omega = 7.292 \times 10^{-5}$ s$^{-1}$ for the Earth.

We do not repeat all the details described in Jánosi et al. (2019), just recall the most important observations. For an isolated Gaussian vortex, the total eddy kinetic energy and enstrophy are finite over an infinite domain of integration (written in 2D polar coordinates):

$$I_{EKE} = \frac{1}{2}\int_0^\infty 2\pi r v^2(r) dr = \frac{g^2 \pi \eta_0^2}{2f^2} \quad , \tag{3}$$

$$I_Z = \frac{1}{2}\int_0^\infty 2\pi r \xi^2(r) dr = \frac{g^2 \pi \eta_0^2}{f^2 R_{eff}^2} \quad . \tag{4}$$

The total eddy kinetic energy integral $I_{EKE}$ depends only on the peak height $\eta_0$ of the vortex at a given geographic latitude. The total enstrophy integral $I_Z$ is very similar, but having an effective radius $R_{eff}$. That is why the ratio of the two integrals is simply

$$\frac{I_{EKE}}{I_Z} = \frac{1}{2}R_{eff}^2 \quad . \tag{5}$$

When the time-series of the two integrals are properly correlated, the ratio provides a key to estimate an effective size for a single "proxy vortex" characterizing the given area of integration.

## 2.3 Analysis of correlations

In order to characterize correlations for the two integrals $I_{EKE}$ and $I_Z$, we determined the Pearson correlation coefficient $P$ by using the standard definition

$$P_{I_{EKE}, I_Z} = \frac{\langle [I_{EKE}(t) - \overline{I_{EKE}(t)}][I_Z(t) - \overline{I_Z(t)}]\rangle_t}{\sigma_{I_{EKE}}\sigma_{I_Z}} \quad , \tag{6}$$

where the overline indicates mean value, the nominator is a temporal mean value of the products and $\sigma$ is the standard deviation. All calculations were performed in a Python environment (version 3.6) with the standard `Numpy` (Harris et al., 2020) and `Scipy` (Virtanen et al., 2020) packages. Maps were drawn by the `Basemap` module (https://matplotlib.org/basemap/, last accessed on 27/04/2022).

Pearson correlation is the most common metric for the evaluation of a linear association between two time-series. An often used alternative metric to test arbitrary but monotonous association between two time-series is provided by the Spearman rank correlation coefficient, which is actually Pearson's correlation coefficient applied to the ranks of the observations. According to our tests, the two coefficients are essentially identical for the integrals and, therefore, we use Pearson's $P$.





Special considerations are required to determine the area of integration. The reason is that practically no correlations exist
between $I_{EKE}$ and $I_Z$ in single grid cells of $0.25° \times 0.25°$. The plausible explanation is that a single velocity vector per
grid cell (assumed to be a characteristic value over the cell) cannot resolve finer structures, therefore the kinetic energy and
enstrophy are changing almost independently in time. We analyzed this question in details in Jánosi et al. (2019). On the global
scale, an analogously detailed analysis would be computationally too excessive, and it does not seem inevitable. Instead, we
will compare results for three different tilings: $21 \times 21$, $11 \times 11$, and $5 \times 5$ grid cells (the odd numbers have the benefit that
the position of the central grid cell defines a clear geographic location for a tile). Close to the shorelines, we kept tiles where a
large enough number or grid cells were over the oceans, specifically: at least 200 out of 441, 80 out of 121, and 20 out of 25,
respectively. The tessellations were constructed with an overlap of 1 grid cell wide stripes at the edges, in order to have an easy
reference to the central coordinates of tiles. In this way the spacing of tile centers is $1.0°$ for the smallest, $2.5°$ for the medium,
and $5.0°$ for the largest tiles.

If the correlations between $I_{EKE}$ and $I_Z$ are strong enough, we can use the simple relationship in Eq. (5) to estimate an
effective radius $R_{eff}$ for a single Gaussian proxy vortex.

### 2.4  Consideration of eddy amplitudes

We noted before that the integrated kinetic energy over an infinite domain Eq. (3) depends only on the squared peak height $\eta_0^2$
of an isolated Gaussian vortex. Theoretically, we can exploit this fact to obtain proxy vortex amplitudes and compare them with
the amplitudes stored in the mesoscale eddy database (Faghmous et al., 2015). In the practice, however, we run into the problem
that eddies are never isolated in the ocean, and an estimation of KE or EKE is performed always over a finite area. Numerical
tests prove that time series of EKE at a given location but at two different tile sizes are quite similar when the area relationship
is properly considered. E.g., the kinetic energy behaves similarly (but not identically) at a fixed [lat,lon] center for integration
areas of $11 \times 11$ and $21 \times 21$ grid cells when the former is multiplied by the area ratio of $441/121 \approx 3.645$. A comparison is
also possible after normalization, the result in this case is energy per unit area (commonly used in oceanography). However,
such normalization does not work in Eq. (3) simply because an increasing area of integration does not yield to energy saturation
in any real surface flow field in contrast to an isolated vortex.

For the above explained reason, we reverse the consideration of Eq. (3), in order to get a hint about the partition of kinetic
energy between geostrophic vortices and the background flow. We assume that the majority of oceanic eddies has a Gaussian
shape and we estimate their total kinetic energy by inserting the measured (squared) amplitudes into Eq. (3) from the vortex
census. Next we compare the sum of kinetic energies for the individual eddies and the total kinetic energy obtained from the
velocity anomaly field. We perform this procedure for the two larger tilings ($21 \times 21$ and $11 \times 11$ grid cells) where we expect
that MEs exist in the given tile and time.

### 2.5  Westward drift of mesoscale eddies

A well-known and widely analyzed feature of eddy trajectories is the general tendency for westward propagation in lack of
strong counter-currents (Cushman-Roisin et al., 1990; Chelton et al., 2007, 2011; Early et al., 2011; Kurian et al., 2011; Drótos



and Tél, 2015; Brach et al., 2018; Cetina-Heredia et al., 2019; van Sebille et al., 2020; Wichmann et al., 2021, just to mention of a few references from the existing vast literature). The usual parsimonious explanation is based on the beta-plane effect: the Coriolis parameter is slightly different on the two opposite sides of an eddy (in the meridional direction). Beta-plane approximation exploits the linear expression $f = f_0 + \beta y$, where $f_0(\varphi)$ is the Coriolis parameter at the reference latitude $\varphi$, $y$

is a (linear) meridional distance, and the slope factor $\beta = df/dy|_\varphi = 2\Omega\cos(\varphi)/R_E$, where $R_E = 6378100$ m is the radius of Earth. Manifestly, $\beta$ is largest around the equator and decreases toward larger latitudes on both hemispheres.

The beta-plane approximation allows an analytical solution with the result that all eddies (cyclonic and anticyclonic) propagate westward, and the speed does not depend on the size and height (or depth) of a vortex obeying geostrophic equilibrium. The simple formula for the drift speed is the same as for long nondispersive Rossby waves (Cushman-Roisin et al., 1990):

$$U_d = -\beta R_d^2 \ , \tag{7}$$

where $R_d = \sqrt{g'H}/f_0^2$ is the radius of deformation with the reduced gravity $g'$ and the mean layer thickness $H$ (outside of an eddy). The latter is usually estimated as the thickness of the mixed layer down to the pycnocline, as a first approximation. Nevertheless, to obtain a precise value for $R_d$ is not a trivial task (see e.g., Nurser and Bacon (2014).) The picture is further complicated by the weak nonlinear effects present in a quasigeostrophic approach resulting in an amplitude (thus traveling

distance) dependence of propagating speeds (see e.g., Figs. 9 and 10 in Early et al., 2011).

We will compare our results with the linear estimate. We used the gridded data set for Rossby radii compiled by Chelton et al. (1998), it is available on the internet (https://ceoas.oregonstate.edu/rossby_radius, last accessed on 27/04/2022).

The appealing aspect of the beta-plane approach is that the drift speed does not depend on the characteristics of individual eddies, although they transport kinetic energy and vorticity westward. We exploit this fact to estimate westward propagation

velocities by evaluating the cross correlation $X(\tau)$ of integrated kinetic energy $I_{EKE}(t)$ between neighboring tiles in the zonal direction:

$$X(\tau) = \frac{\langle[I_{EKE}(t)_i - \overline{I_{EKE}{}_i}][I_{EKE}(t-\tau)_{i-1} - \overline{I_{EKE}{}_{i-1}}]\rangle}{\sigma_i\sigma_{i-1}} \ , \tag{8}$$

where the time lag $\tau$ represents a temporal shift between the two time-series by $\tau$ days, overbar denotes temporal mean, and $\sigma$ is the standard deviation of $I_{EKE}(t)$ in the given tile.

For the validation, we explored again the eddy census by Faghmous et al. (2015). Besides the collection of individual MEs, they provide 2,758,222 eddy tracks (in a single giant text file containing 36,662,978 lines and 7 records by line). The temporal resolution of tracks is 1 day (AVISO standard). Since the cross correlation method (8) outlined in the previous paragraph detects only zonal drifts, we extracted the same information (mean zonal drift speed as a function of the latitude) from the available eddy tracks. Daily travel distances in the zonal direction were determined by the widely used haversine formula (Cotter, 1974),

where the mean value of the latitudes at day$_i$ and day$_{i+1}$ and the zonal distance of longitudes were inserted.





# 3   Results and discussions

## 3.1   Correlations between $I_{EKE}(t)$ and $I_Z(t)$, vortex radii

The geographic distribution of the Pearson correlation coefficient $P$ for $I_{EKE}(t)$ and $I_Z(t)$ is shown in Figs. 2a,b,c for the three

tilings, respectively. The common feature is the strongly decreasing correlation in the latitude band [10°S - 10°N] around the

equator. This behavior is expected because the Coriolis effect vanishes at the equator, thus geostrophic eddies have very short

survival time when drifted close to the zero latitude. The color coding in Fig. 2 indicates stronger correlations for larger tile

sizes, which is in full agreement with the result in Jánosi et al. (2019). Somewhat surprising is the fact that strong correlations

between $I_{EKE}(t)$ and $I_Z(t)$ appear in several ocean basins at the smallest tile size of $1.25° \times 1.25°$, mostly at the eastern

boundaries of the basins (south to Australia and around the Agulhas-, Antarctic Circumpolar-, Gulf- and Kuroshio Current).

The yellowish colors dominating Fig. 2c represent lower but statistically significant correlations with $P$ values between 0.6

and 0.7.

The strong correlations permit to give an estimate for an effective proxy vortex radius $R_{eff}$ with Eq. (5) as $R_{eff} = \sqrt{2I_{EKE}/I_Z}$. For simplicity, we show here only the map for the finest tiling in Fig. 3, adding however, that all three pan-

els look very similar. Likewise in Fig. 2, we observe anomalies in the equatorial band of [10°S - 10°N] as unusually large

vortex sizes.

For the validation of the results, Fig. 4 exhibits a comparison for five data-sets. From the vortex census by Faghmous et al.

(2015), we used the parameter of eddy area for each individual case, because it is given in units of km² and not in grid cell

number, therefore we did not need to determine the meridional corrections. We determined and plotted zonal means for an

equivalent radius $R_{equiv}$ which is a radius of a circular vortex of the same area as recorded. Since the identified eddies were

fitted with ellipses by Faghmous et al. (2015), the lengths of the major and minor axes $a$ and $b$ were recorded. We checked

the statistics by determining the usual parameter of eccentricity $\epsilon = \sqrt{1 - (b/a)^2}$. Characteristic zonal mean values are around

$\epsilon \sim 0.75$ ($b \approx 0.66a$) in the large ocean basins between latitudes [40°S - 40°N]. Northward and southward of this band the

eccentricity gradually increases close to 1 (strongly elongated shapes). This result is in agreement with Chen et al. (2019)

where statistics of fitted ellipses and their orientations is presented for 2.6 million individual eddies. However, a simple visual

check of a couple of maps (not shown here) indicates that an ellipsoidal fit is also an approximation. Mesoscale eddies are

strongly interacting with the neighbors, and before they decay, the isocontours of sea height anomalies exhibit rather complex

features with lobes and waves around.

The agreement between the five estimates of zonal mean effective radii $R_{eff}$ and $R_{equiv}$ is rather satisfactory for the

latitudinal bands 80°S - 20°S and 20°N - 80°N. Gray band indicates the equatorial zone where we obtained poor correlations

(see Figs. 2). Despite this fact, we plotted the estimates for the following reason. The vortex data bank (Faghmous et al., 2015)

contains a surprisingly large number of mesoscale eddies in this "gray zone" (see Fig. 4) with rather large equivalent radii.

Since this band cannot be a source of geostrophic eddies, they must be advected here from higher and lower latitudes. These

eddies have systematically low amplitudes and, therefore, their detection by isocontours can have a large error and can result




in easily underestimates of their size. Unfortunately, eddy radius estimates by Eq. (5) are rather unreliable in the gray zone too,
because of the low level of correlations at each tiling (c.f., Figs. 2).

Note that averaging should be performed with care. This is due to a finite size effect: it occurs often that no eddies (more precisely, no eddy centers) are identified in a given day and in a given tile by the eddy census. Actually, from the aggregated data set of size 7792×36×72 (days×long×lat) 26.3% for the largest tiles, and 31.1% for the medium size tiles (out of 7792×72×144) are zeros. For this reason, all zero values in the tile-wise eddy census were masked and the mask was used in
all data series before computing mean values.

The dashed line in Fig. 4 denotes the mean Rossby radius of deformation (Chelton et al., 1998). It is well known that MEs have usually larger equivalent radii than the Rossby radius at a given latitude due to essential (albeit weak) nonlinearities (Matsuura and Yamagata, 1982; Roisin and Tang, 1990; Dewar and Killworth, 1995; Willett et al., 2006; Chelton et al., 2011), and it is properly reproduced in high resolution ocean models (Ajayi et al., 2020; Moreton et al., 2020). The agreement between
the proxy vortex approximation and a previous global direct vortex census by Chelton et al. (2011, c.f., Fig. 12) is remarkably good.

### 3.2    Vortex amplitudes and kinetic energies

In order to compare amplitudes of proxy vortices and amplitudes (stored height parameters) in the eddy census data base, one can explore Eq. (3). Since $I_{EKE}$ depends only on the squared peak height of a Gaussian proxy vortex (and some constants),
proxy vortex amplitudes can be obtained by inserting the sum of kinetic energies belonging to identified eddies $\iint EKE_{ec}$ in the eddy census. As described in Subsection 2.4, we performed the calculation in the reverse direction. Instead of attempting to extract sum of squared peak heights from $I_{EKE}$, we estimated tile-wise the eddy kinetic energy from the eddy census $\iint EKE_{ec}$ by inserting the stored amplitudes into Eq. (3). This approximation assumes that MEs have Gaussian sea level anomaly profiles, in general. The result for temporal mean values is illustrated in Fig. 5. As expected, higher integrated $EKE_{ec}$
values are characteristic in the vicinity of energetic ocean currents, where a large number of vortices are generated by baroclinic instabilities at the current edges (e.g., Willett et al., 2006; Smith, 2007; Badin et al., 2009; Molemaker et al., 2015; Marta and Isachsen, 2018; Pnyushkov et al., 2018).

Figure 6 illustrates the tile-wise mean ratio $I_{EKE}/\iint EKE_{ec}$ which is equivalent with the ratio of squared amplitudes $\eta_0^2/\eta_{ec}^2$ [see Eq. (3)] in the Gaussian approximation. The map in Fig. 6b is rather pixelated indicating high variabilities. Indeed,
both in space and time the local values of the eddy kinetic energy ratios fluctuate strongly between 0.5 and 15. Note that here again, only the dates were considered in the determination of mean values where at least a single eddy center was detected in a given tile. The range of mean ratios is in the same order of magnitude as found by Amores et al. (2018) (they reported on a partition ratio between 1 and 5 fluctuating strongly in time), however it differs from Fig. 5b in Jánosi et al. (2019). In the later, a mean ratio around 2 was deduced in a restricted ocean region along the shoreline of Oregon and California. However, the
way of the estimation of eddy heights was different. While in Jánosi et al. (2019) the values of sea level anomaly at the very center of identified eddies were used as proxies, here we directly extracted the amplitude parameters from the eddy census by Faghmous et al. (2015).





This statistics also suffers from an additional finite size effect, besides the occasional lack of identified vortices. When the location of an ME is close to the boundary of the given tile, its contribution to $\iint EKE_{ec}$ is larger than to $I_{EKE}$ (determined
by direct counting from geostrophic velocity anomalies $[u'_g, v'_g]$). This is because the implied integration domain in Eq. (3) is infinite for a given eddy, but only about half of it contributes to the counting of total kinetic energy inside the tile. As a consequence, in several cases the daily ratio is smaller than one, suggesting that eddy centers in the given tile are close to the tile edges. Fortunately this bias is not too strong, at least the zonal mean values shown in Fig. 6a are consistent for the medium and largest tile sizes (far enough from the equator).

An essential and well known source of input data errors is the difficulty of measuring eddy amplitude from altimeter data. Most of the recorded amplitudes are small in the range of a few centimeters, and the reference level is usually the approximate height of the identified close contour (depending on the method). Small eddies (both in extent and height) are poorly resolvable at the available spatial resolutions, therefore estimates are certainly loaded with errors. In order to check the sensitivity of the method to amplitude errors, we repeated all the calculations where the amplitude values are systematically shifted up by +1
cm (note that large eddies of half a meter or similar are hardly affected by such a shift). The results changed rather strongly: while the curves of zonal mean values keep all the presented geographic tendencies (see Fig. 6a), the partition ratio dropped to a mean value of 4.3 instead of 6.2, illustrating the sensitivity of our approach.

### 3.3 Eddy drift properties

The results for the analysis of (mostly) westward drift (see Subsection 2.5) are presented in Figs. 7 and 8. The geographic
distribution for the coefficient of cross-correlations [$X(\tau)$ in Eq. (8)] in Fig. 7a exhibits high statistical significance almost everywhere, particularly in the band [30°S - 30°N]. The time lags $\tau$ at the maximum cross-correlations obey definite negative values in this band (blue colors in Fig. 7b) indicating marked westward drift. Eastward drift (red colors in Fig. 7b) is characteristic in the regions of the Antarctic Circumpolar Current, at the northern Gulf- and Kuroshio Currents, as expected. Note the appearance of large white areas along the equator and at larger than 40° latitudes on both hemispheres indicating very short
time lags of 1-2 day or even zero (recall that the temporal resolution of flow fields is 1 day). Near zero lags result in anomalously large values at velocity estimates as distance over time lag. By checking several flow field maps, it seems that these regions can be characterized by weak vortex activity. One plausible explanation is that lag zero significant cross-correlations are related to simultaneous kinetic energy changes where the wind field over extended areas pumping kinetic energy into or from the oceanic surface layer.

Fig. 8 illustrates the comparison of drift speed results, zonal mean values for the the vortex data bank (red symbols and orange error bars, Fig. 8a) and for the cross-correlation analysis (blue, Fig. 8b). The results of the cross-correlation method are in a good agreement with vortex tracking statistics in the latitude bands 5°–30° on both hemispheres. Stronger discrepancies are present in the bands 30°–50°, again on both hemispheres. As for the narrow band 5°S–5°N along the Equator, the vortex census statistics indicates a sharp drop of propagation speeds. Since here mesoscale eddies cannot survive a long time in the
lack of Coriolis effect, the statistics breaks down.





The observed discrepancies can be explained by the fact that mesoscale eddies are not the only form of kinetic energy transport on the ocean surfaces. We have already noted that the eddy kinetic energy field estimated from the geostrophic velocity anomalies (see Fig. 1a) has a surprisingly large contribution from the energetic current systems of apparently strong temporal and spatial variabilities. The strong and narrow eastward currents at higher latitudes are parts of the top branch of

ocean gyres. Probably this explains the anomalies present also in the vortex tracking statistics: particularly in the latitude bands 40°–50° on both hemispheres, the mean propagating velocity is zero or near zero (see Fig. 8). Standing mesoscale eddies can exist only where the westward drift is balanced by an eastward background flow. Along the equator, freely propagating, nondispersive, linear, first-mode baroclinic Rossby waves are common (Chelton and Schlax, 1996, see also Fig. 5A) suggesting that cross-correlation analysis detects such Rossby waves in this band. Furthermore, long living MEs might be related to Rossby

waves as suggested recently by Sutyrin et al. (2021). When steady propagating baroclinic vortices are embedded in a large-scale vertical shear, they can radiate Rossby waves without decay by extracting available potential energy from the vertically sheared background flow (Sutyrin et al., 2021).

### 3.4 Eddy kinetic energy transport

In order to further elaborate the possible reasons of the discrepancies between the drift speed values obtained by the two

methods, we studied the Hovmöller diagrams (time - longitude plots) for several latitudes. Three examples are illustrated in Fig. 9. Red coloring indicates the time evolution of the locations where the eddy kinetic energy per grid cell has relatively high values. In Fig. 9a, the visible drift is eastward, no wonder, because this region falls into the band of Antarctic Circumpolar Current (see Fig. 7b). The pattern suggest that vortical activity is patchy formed by small eddies of short lifetimes. The usual westward propagation pattern is illustrated in Fig. 9b. Notably, this eastern section of the north Pacific basin along 31°N is

also interrupted by white areas of short time lags (see Fig. 7b) indicating the lack of mesoscale eddies. Even after the filtering, $\tau \geq 3$ lags give anomalously large drift speed values to the zonal mean. An interesting situation is exhibited in Fig. 9c. This section along the Aleutian Islands in the northeastern Pacific is characterized by the quasiperiodic appearance of a single giant energetic eddy or ocean ring advected by the Alaskan Stream. Such an event is depicted in Fig. 10, where the huge ring of a diameter 300-350 km passes by westward between day 1500 (1997-02-09) and day 2000 (1998-06-24). Similar giant rings are

known and studied mostly by satellite altimetry for a few decades (Ladd et al., 2007; Ueno et al., 2009; Lyman and Johnson, 2015; Prants et al., 2019). Recent results suggest that their unusually long lifetime is related to the bottom topography (Gulliver and Radko, 2022).

### 4 Summary

Since we already discussed the results in the Subsections above, here we list the main findings of this work.

– The eddy kinetic energy (EKE) obtained from the 2D geostrophic velocity anomalies $[u'_g, v'_g]$ of the AVISO data bank partly reflects the activity of mesoscale eddies, however it has significant contributions from the meandering and swinging of the major western boundary and equatorial currents (see Fig. 1).



- The time dependent integrated eddy kinetic energy $I_{EKE}$ and integrated enstrophy $I_Z$ are strongly correlated except along the equator. The larger the area of integration the stronger the temporal Pearson correlation (see Fig. 2). While there is no correlation in a single grid cell of size $0.25° \times 0.25°$ with a single mean velocity (assigned to the center of grid cells), and integration area of $5 \times 5$ grid cells ($1.25° \times 1.25°$) is enough to exhibit significant correlations almost everywhere (see Fig. 2c).

- The effective zonal mean radii of proxy vortices $R_{eff}$ obtained by Eq. (5) are in good agreement with zonal mean values extracted from the eddy census by Faghmous et al. (2015) away from the equatorial band [$10°$S, $10°$N] (see Fig. 4). Here correlations between $I_{EKE}$ and $I_Z$ are decaying, thus the approximation by Eq. (5) breaks down.

- Integrated eddy kinetic energies are obtained by two estimates. Firstly, $I_{EKE}$ is computed from geostrophic velocity anomalies by direct counting. Secondly, for an approximation of the "true" eddy contribution $\iint E_{ec}$, squared amplitudes from the eddy census by Faghmous et al. (2015) are inserted into Eq. (3). The zonal mean ratio of $I_{EKE} / \iint E_{ec}$ is somewhat larger than found in previous analyses (see Fig. 6a), however this ratio is very sensitive to small errors on recorded eddy amplitudes. A uniform upward shift by 1 cm resulted in a $\sim 30$ % drop in zonal mean values.

- As for the propagation of kinetic energy and drift of eddies, the estimates from time lagged cross-correlation analysis of EKE fields are in general agreement with vortex tracking statistics considering order of magnitudes and sign (see Fig. 8). However, it provides somewhat larger drift velocities than that of individual mesoscale eddies. Discrepancies are not unexpected because besides mesoscale eddies, Rossby wave activities, coherent currents, and other propagating features on the ocean surface apparently contribute to the zonal transport of kinetic energy.

We can shortly conclude that the original proposal by Jánosi et al. (2019) works on a global scale. The comparison with the rich eddy census data base by Faghmous et al. (2015) resulted in good agreement in many aspects validating the Gaussian proxy vortex approach. Apparent discrepancies provide also useful insight e.g. into the separation of kinetic energy between "true" eddy and background flow contributions, or into the mechanisms of kinetic energy transport.

*Code and data availability.* Global geostrophic velocity fields are openly available after registration at the E.U. Copernicus Marine Service (https://resources.marine.copernicus.eu/products). The mesoscale vortex data bank assembled by Faghmous et al. (2015) is available at https://datadryad.org/stash/dataset/doi:10.5061/dryad.gp40h, together with Matlab routines in the Zenodo source-code repository (https://zenodo.org/record/13037#.YV15T-e8phE). All the other calculations and plotting are based on standard Python modules described in Section 2 (Data and methods) in details.

*Author contributions.* I.M.J. designed research; I.M.J. and M.V. performed research; H.K. and J.A.C.G. contributed new numerical/analytical tools; I.M.J. and H.K. analyzed data; and I.M.J., H.K., J.A.C.G. and M.V. wrote the paper.



*Competing interests.* The authors declare no conflict of interest.

*Acknowledgements.* This work was supported by the Hungarian National Research, Development and Innovation Office under grant numbers FK-125024 and K-125171, and by the Visitor Programme of Max Planck Institute for the Physics of Complex Systems. J.A.C.G. was
supported by CNPq, Brazil, grant PQ-305305/2020-4.



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



**Figure 2.** Pearson correlation coefficients $P$ for the integrated kinetic energy $I_{EKE}(t)$ and enstrophy $I_Z(t)$ at three different tilings determined for the whole period of Aviso records. **(a)** $21 \times 21$ grid cells; **(b)** $11 \times 11$ grid cells; **(c)** $5 \times 5$ grid cells. (For details see Subsection 2.3 .)



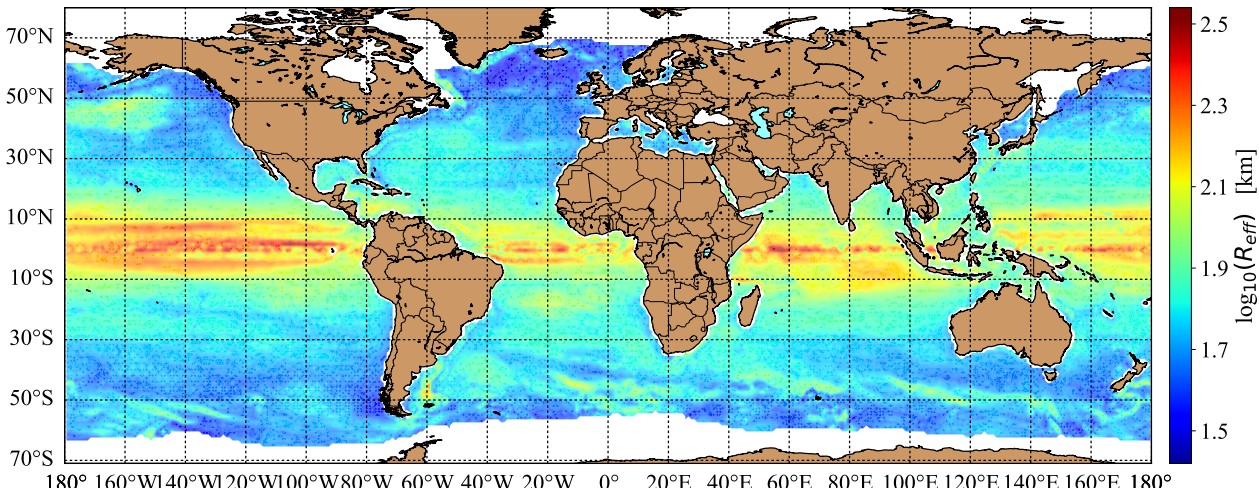

**Figure 3.** Geographic distribution of the effective radius $R_{eff}$ based on the estimate by Eq. (5) for the finest tiling of $1.25° \times 1.25°$. The color scale is logarithmic for a better visualization. White pixels denote seasonally ice covered regions.

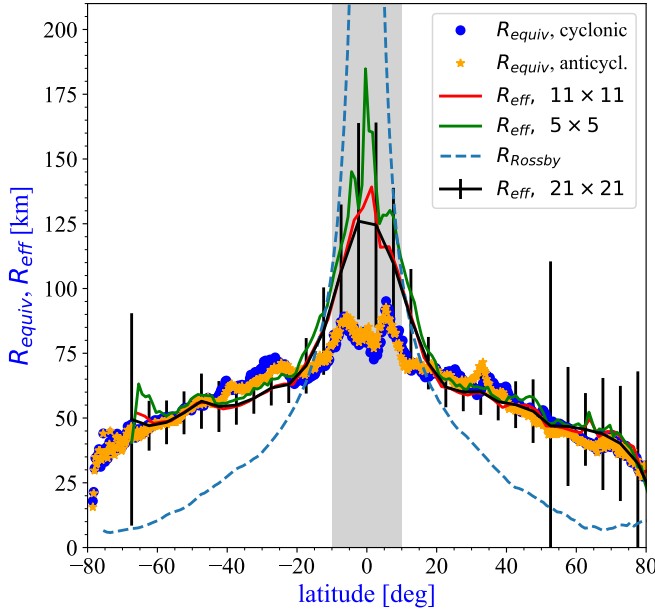

**Figure 4.** Zonal and temporal mean values of the characteristic vortex sizes $R_{equiv}$ and $R_{eff}$ for five data sets. Solid symbols are statistics from the vortex data base (Faghmous et al., 2015), 32 millions of cyclonic (blue symbols) and about the same number of anticyclonic (orange stars) MEs. Solid lines are the estimates by Eq. (5) for the three tilings, see legends. Gray band denotes the equatorial region of low correlations (see Fig. 1) where the estimates for $R_{eff}$ are particularly unreliable. Black error bars are for $R_{eff}$ statistics ($1\sigma$) with the largest tiles ($21 \times 21$) of integration. For a comparison, dashed line indicates the zonal mean Rossby radius of deformation, data from Chelton et al. (1998).



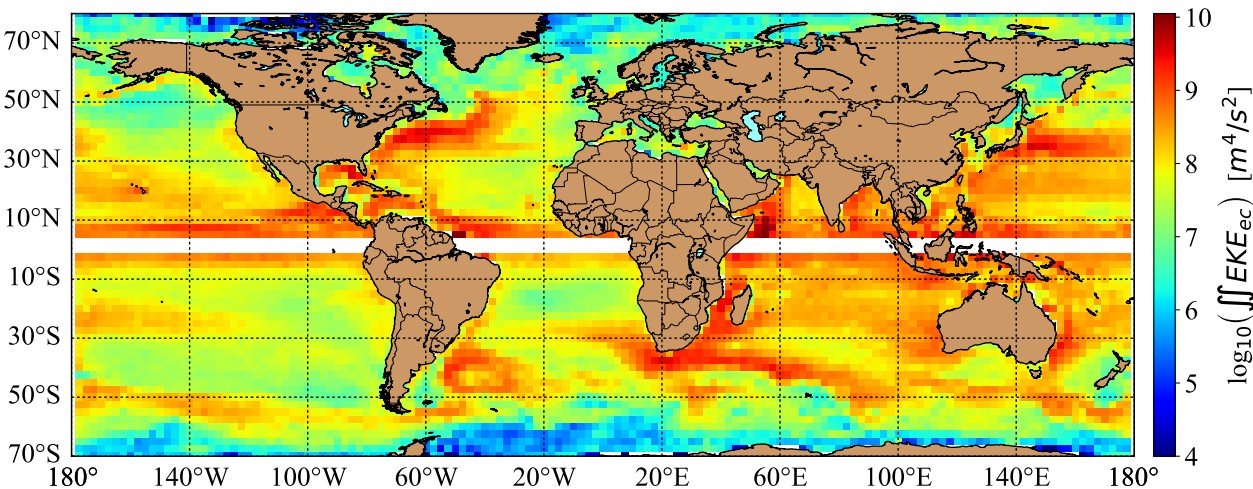

**Figure 5.** Temporal mean eddy kinetic energy $\iint EKE_{ec}$ estimated from the squared amplitude parameters $\eta_{ec}^2$ of the eddy census by Eq. (3) for the tiling of $11 \times 11$ grid cells. The white band indicates the equatorial region where estimates diverge, because the Coriolis parameter $\sim \sin(\varphi)$ in the denominator of Eq. (3) tends to zero. The color scale is logarithmic for a better visualization.

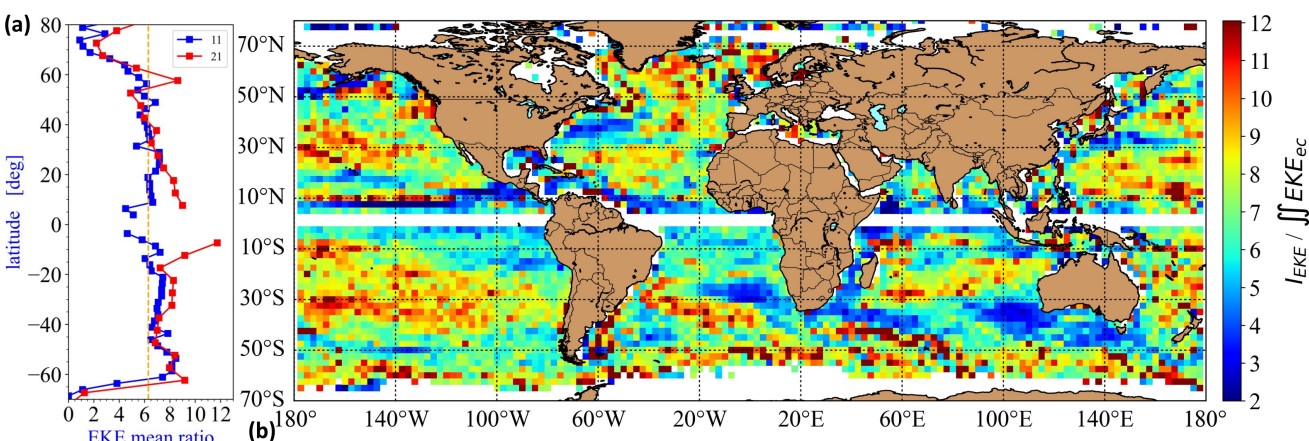

**Figure 6.** Temporal mean eddy kinetic energy ratio $I_{EKE}/\iint EKE_{ec}$, the later is estimated from the squared amplitude parameters $\eta_{ec}^2$ of the eddy census by Eq. (3). **(a)** Zonal mean values for the tilings of $11 \times 11$ (blue) and $21 \times 21$ (red) grid cells. The vertical dashed line (orange) is not fitted, it serves only for orientation. **(b)** Geographic distribution for the tiling of $11 \times 11$ grid cells. The white band indicates the equatorial region where estimates diverge, because the Coriolis parameter $\sim \sin(\varphi)$ in the denominator of Eq. (3) tends to zero. The color scale is linear.

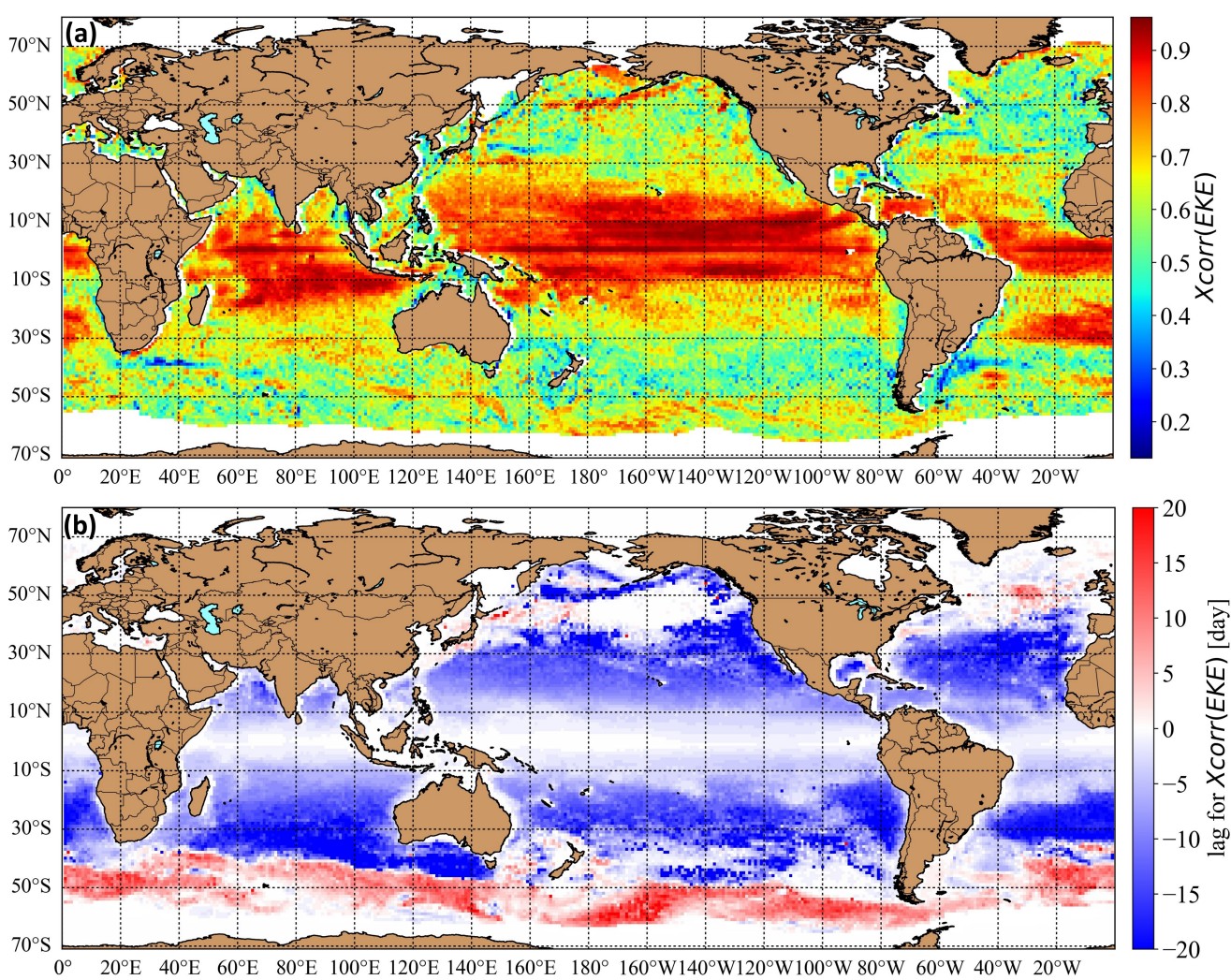

**Figure 7.** Time dependent cross-correlation analysis of the EKE field by Eq. (8) for the smallest tiling (5 × 5 grid cells). Only zonal neighbors are considered. The color scales are linear. **(a)** The level of maximal cross-correlation for each valid tile. **(b)** The time lag of maximal cross-correlation for each valid tile. Blueish/reddish coloring indicates dominant westward/eastward zonal drifting tendency.

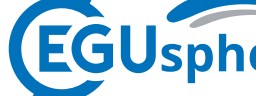



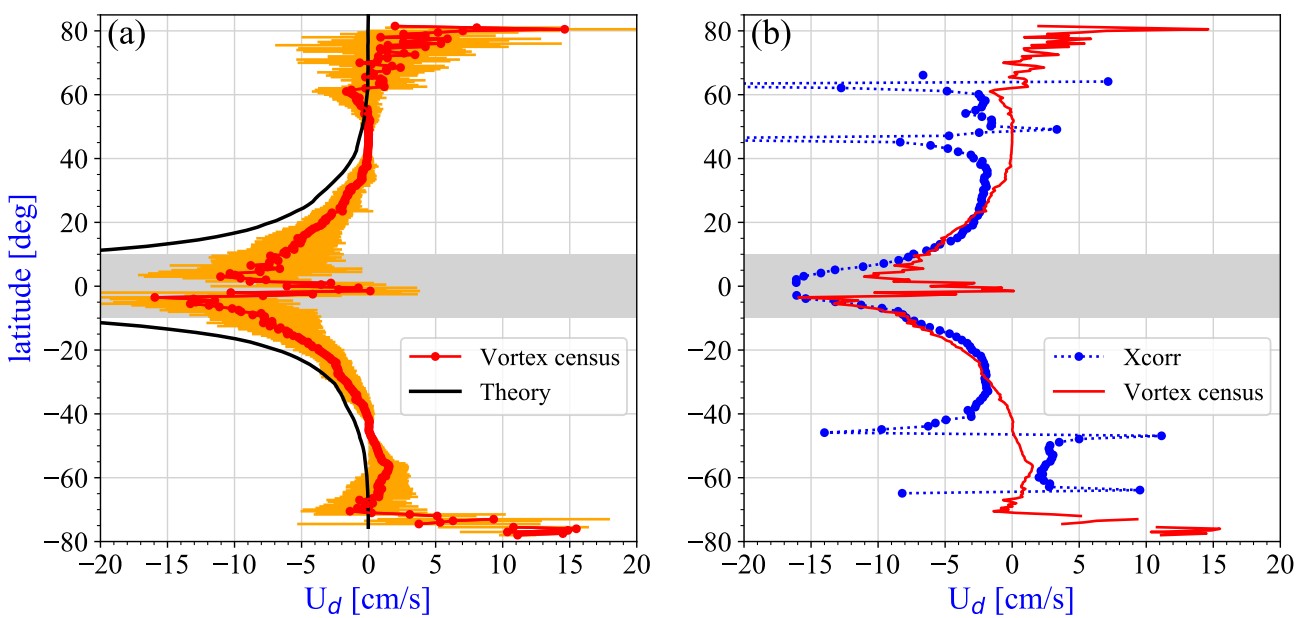

**Figure 8.** Zonal mean values of the characteristic zonal drift speed $U_d$ in units of [cm· s$^{-1}$] for three data sets. Gray bands emphasize the tropical region of particularly strong cross-correlation values (c.f. Fig. 7a). **(a)** Red symbols (with orange error bars) indicate the result from the tracks of vortex census. Black solid line is the estimate for nondispersive Rossby waves Eq. (7). **(b)** Blue symbols denote the result for the cross-correlation analysis at the smallest tiling ($1.25° \times 1.25°$), anomalously low time lag values ($\tau = 0, 1, 2$ days) are filtered out from the statistics. Red line is the mean drift speeds from (a).



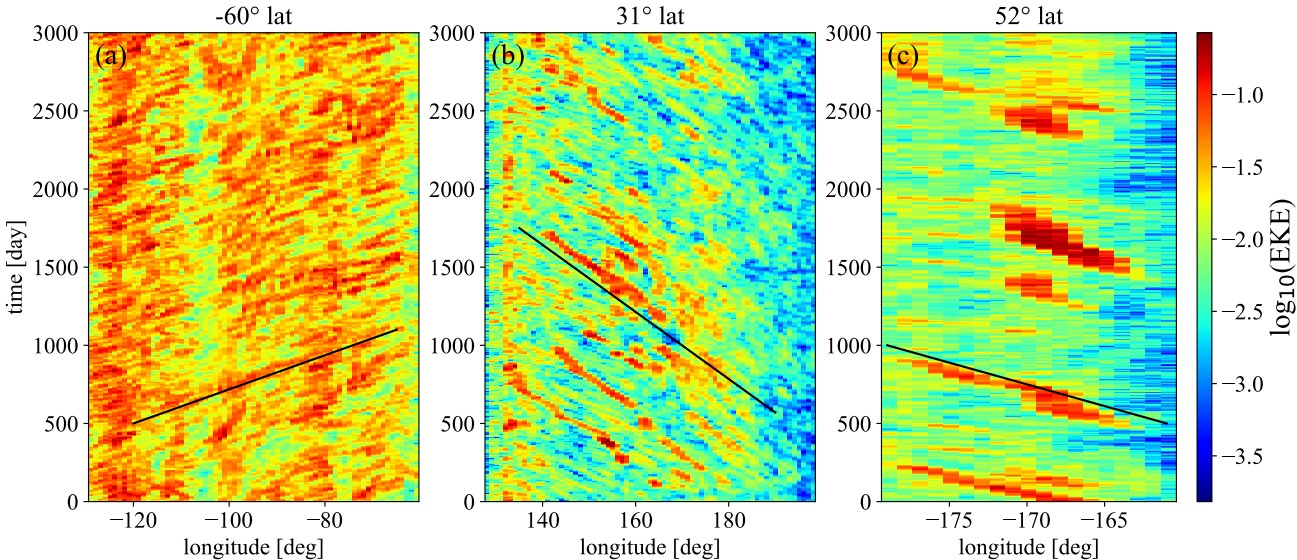

**Figure 9.** Hovmöller plots (time - longitude diagrams) of eddy kinetic energy propagation for three particular regions. Latitudes are indicated in the title of panels. Black lines guide the eye for characteristic slopes. Color scales are logarithmic. **(a)** A southern Pacific section in the region of Antarctic Circumpolar Current at the latitude $60°$S. The eastward drift speed (black line) is estimated as 5.8 cms$^{-1}$. **(b)** A northern Pacific section at the latitude $31°$N. The westward drift speed (black line) is around -5.2 cms$^{-1}$. **(c)** A northern Pacific section at the latitude of $52°$N, rather close to the Fox Islands. The rare extreme energetic vortices obey a westward drift speed around -2.9 cms$^{-1}$.



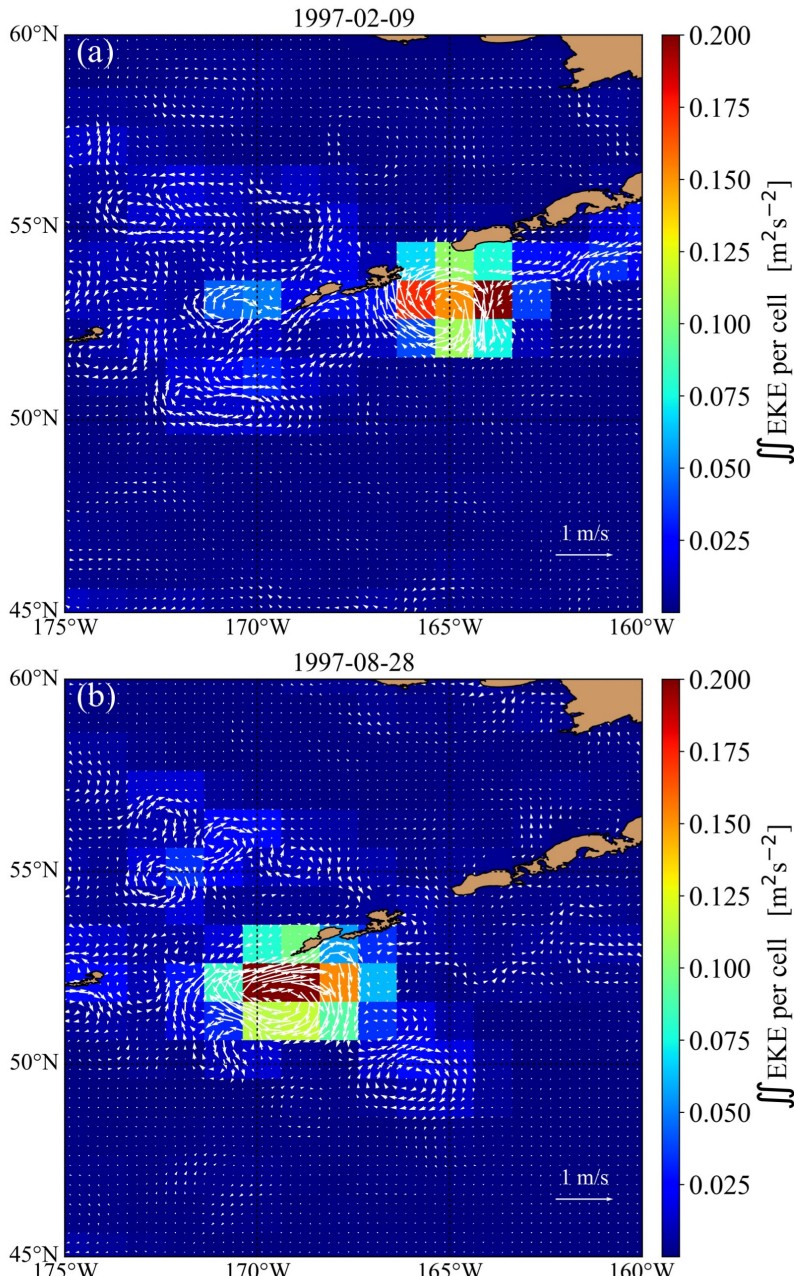

**Figure 10.** Snapshots of the geostrophic flow field and integrated kinetic energy (color scale is linear) for two time instances in the northern Pacific region analyzed in Fig. 9c. **(a)** 1997-02-09 (day 1500 in Fig. 9c). **(b)** 200 days later, 1997-08-28 (day 1700 in Fig. 9c). The radius of the huge anticyclonic vortex has a length around 300-350 km (note that the apparent elongation is the consequence of equidistant cylindrical map projection).