# Peer review of "Global coarse grained mesoscale eddy statistics based on integrated kinetic energy and enstrophy correlations"

_EGUsphere, 2022_

## Author Comment (AC1)

MS No.: egusphere-2022-231

**Response to Referee #1**

We are grateful for the comments of the referee. We appreciate very much the time and effort which the referee has devoted to our manuscript. The report will enable us to improve the presentation of the material and to put it into the right context. Here we list all the critical remarks (in italics) and give an explanation of our points.

General evaluation:

*RC1: .... I recommend their manuscript for publication with only minor comments listed below.*

**Response**:
Many thanks for the supporting opinion.

Minor comments:

*RC1: Sections 2.3 and 3.1: It would be interesting to have a physics-based discussion on why one would expect a temporal correlation between the area-integrated eddy kinetic energy and enstrophy.*

**Response**: As described by Hopfinger and van Heijst (1993, page 244), for two-dimensional flow it is convenient to describe the motion in terms of the vorticity equation, which then takes a scalar form. In the case of flow in a horizontal plane with velocity $\mathbf{v} = (u, v, 0)$ the vorticity $\omega = \omega \mathbf{k}$ is given by:

$$\frac{\partial \omega}{\partial t} + J(\omega, \Psi) = \nu \nabla^2 \omega \ , \tag{1}$$

where $J$ is the Jacobian operator and $\Psi$ denotes the stream function, defined according to $\mathbf{v} = -\mathbf{k} \times \nabla \Psi$. By definition $\Psi$ satisfies $\nabla^2 \Psi = -\omega$. Note that $\omega$ represents here the absolute vorticity, which is equal to the sum of the relative vorticity and the planetary vorticity: $\omega = \omega_{rel} + f$. From the vorticity equation (1) it is inferred that steady inviscid flows are described by $J(\omega, \Psi) = 0$, and this implies that $\omega = F(\Psi)$, with $F$ *any* integrable function.

Hopfinger and van Heijst (1993) then discuss several known models for discrete vortex flows, such as Rankine, Oseen, Lamb, and Burgers vortices. A relatively simple model describing isolated vortex structures with continuous vorticity and velocity distributions was used by Carton and Mcwilliams (1989). The expressions for nondimensional tangential velocities $v(r)$ and vorticities $\omega(r)$ (where normalizations are performed by appropriate length and velocity scales $R$ and $V$):

$$v(r) = \tfrac{1}{2} r \, \exp(-r^q) \ , \tag{2}$$

$$\omega(r) = \left(1 - \tfrac{1}{2} q r^q\right) \exp(-r^q) \ . \tag{3}$$

Here $q$ is the so-called steepness parameter that controls the shape of the profiles. An appealing property of forms (2) and (3) that the squares of both have finite values of the spatial integrals over infinite domains for any $q > 0$ integer or non-integer values, usually determined by the $\Gamma$ (Gamma) function. Consequently, the ratios of the spatial integrals are also finite rational numbers. The particular value of $q = 2$ defines the Gaussian vortices.

As for the real ocean, mesoscale eddies are close to Gaussian vortices, according to many analyses cited in the text. A finite area of integration is not closed, eddies are coming and going, emerging and decaying, still when the Gaussian hypothesis holds, then a strong correlation is expected between integrated kinetic energy and enstrophy.

Since all these information is described e.g. in Hopfinger and van Heijst (1993), we inserted a related remark (a short version of this explanation) into the text of Subsection 2.2 ("Gaussian mesoscale eddies").

*RC1: Equations 6 and 8: The notation of the mean of IEKE and IZ are denoted with an overbar while the temporal mean of the nominator is in angle brackets. I would suggest unifying the notation one way or the other for representing the mean.*

**Response**: Thanks, we followed your recommendation and denote all temporal mean values by angle brackets.

*RC1: Line 243: In the later -> latter.*

**Response**: Thanks, we preformed the correction.

*RC1: Lines 322-323: The notation Eec is EKEec in Figure 6. Please unify the notation.*

**Response**: Thanks, we preformed the correction.

**References**

Carton, X. and Mcwilliams, J.: Barotropic and baroclinic instabilities of axisymmetric vortices in a quasigeostrophic model, in: Mesoscale/Synoptic Coherent structures in Geophysical Turbulence, edited by Nihoul, J. and Jamart, B., vol. 50 of *Elsevier Oceanography Series*, pp. 225–244, Elsevier, https://doi.org/10.1016/S0422-9894(08)70188-0, 1989.

Hopfinger, E. J. and van Heijst, G. J. F.: Vortices in rotating fluids, Annu. Rev. Fluid Mech., 25, 241–289, https://doi.org/10.1146/annurev.fl.25.010193.001325, 1993.

---

## Author Comment (AC2)

MS No.: egusphere-2022-231

**Response to Referee #2**

We are grateful for the comments of the referee. We appreciate very much the time and effort which the referee has devoted to our manuscript. The report will enable us to improve the presentation of the material and to put it into the right context. Here we list all the critical remarks and questions (in italics) and give an explanation of our points.

General evaluation:

*RC2: I am generally favorable of the work but have a few comments and questions that I hope the authors can address.*

**Response**: Many thanks for the supporting opinion.

Comments and questions:

*RC2: In the abstract, please consider replacing "super vortex proxy" with "vortex proxy." As you mention in the text, the word "super" may be an overstatement.*

**Response**: Thanks, we deleted the term "super" from the abstract.

*RC2: In Fig. 1, it is unclear from the caption if the quantity being visualized in (b) is $|v_g - v'_g|^2$ or $|v_g|^2 - |v'_g|^2$. Please be more explicit.*

**Response**: Thanks for this question, we reformulate the caption for Fig. 1. Actually, as described in Subsection 2.1, the AVISO data sets contain two velocities. The "raw" geostrophic velocities $[u_g, v_g]$ are determined directly from the measured SSH (sea surface height) data. Velocity anomalies $[u'_g, v'_g]$ are determined as deviations from the long term mean values over the period of 1993-2012 for each grid cell. The latter is assumed to represent short term anomalies related e.g. to drifting mesoscale eddies. The first kinetic energy snapshot (Fig. 1a) is determined directly from $[u'_g, v'_g]$ data as $\frac{1}{2}(u'^2_g + v'^2_g)$ and called eddy kinetic energy. The second snapshot (Fig. 1b) is determined as the difference $\left[\frac{1}{2}(u^2_g + v^2_g) - \frac{1}{2}(u'^2_g + v'^2_g)\right]$. This appears now explicitly under Fig. 1.

*RC2: On line 74, you compare the results to the dataset of Faghmous (2015). Is there a reason? Have you considered also using Chelton et al dataset? Can you please comment in the paper? Would doing so constitute too much additional work?*

**Response**: We do not know precisely which is the Chelton et al dataset you refer to. There is a continuously growing set of data repositories, most of them are based on AVISO altimetry. We recently do aware of the recent development by Pegliasco et al. (2022), which lists several alternatives (Chelton et al., 2007, 2011; Faghmous et al., 2015; Martínez-Moreno et al., 2019; Tian et al., 2020; Zhang et al., 2013). The first global database was presented in Chelton et al. (2011), covering the 1993–2008 period, and it was regularly updated until 2016. There is no any discrimination in our choice, one point was the large number of eddies recorded by Faghmous et al. (2015), plus the easy access and transparent data format. We will certainly use the newest data sets in the future, but to repeat everything takes certainly a longer time and more efforts. Furthermore, since the detecting algorithms are very similar, we do not expect any new added value by using different data set(s) in our coarse grained analysis. We inserted a related comment into the text.

*RC2: In eqs. 1,2,3,4, do you use absolute or anomalous values? I suspect you are using SLA, but it is confusing when you use $v'_g$ to represent anomalies in Fig. 1 and $v_g$ (sometimes $v$, without subscript) to represent the same thing in the text and equations.*

**Response**: There is no velocity in Eq. (1), and we do not analyse SLA in this work. As for (2), (3) and (4), these are basic mathematical forms only. During the subsequent calculations, we use systematically velocity anomalies, because these are suspected in relation with mesoscale eddies. Nevertheless Fig. 1a illustrates that the so called eddy kinetic energy contain many contributions from the steady currents or their fluctuations around the mean. To be more precise, we indicite in Eqs. (2) and (3) that we are using velocity anomalies, and vorticities computed from them.

*RC2: In eq. 5, you essentially define $R_{eff}$ as the ratio of the EKE to Z. But in eq. 2, R is a parameter representing the radius of the eddy. Can you please comment on the relation between $R_{eff}$ and R?*

**Response**: Thanks for pointing out this deficiency. These two parameters are equivalent in the case of an isolated vortex. However, when we use the spatially integrated kinetic energy and enstrophy ratios over the ocean, the integrals can belong to a couple of eddies. There is no any a priori argument why the real radius characterizing the shape of an isolated eddy should be closely related to an $R_{eff}$ parameter related to several eddies (apart from the dimension). We inserted this remark into the text below Eq. (5).

***RC2****: Line 122, the word "inevitable" is perhaps better replaced with another word? I could not understand the sentence.*

**Response**: Thanks for the correction. We changed the word, now the sentence read as: "On the global scale, an analogously detailed analysis would be computationally too excessive, and it does not seem unavoidable."

**References**

Chelton, D. B., Schlax, M. G., Samelson, R. M., and de Szoeke, R. A.: Global observations of large oceanic eddies, Geophys. Res. Lett., 34, L15 606, https://doi.org/10.1029/2007GL030812, 2007.

Chelton, D. B., Schlax, M. G., and Samelson, R. M.: Global observations of nonlinear mesoscale eddies, Prog. Oceanogr., 91, 167–216, https://doi.org/10.1016/j.pocean.2011.01.002, 2011.

Faghmous, J., Frenger, I., Yao, Y., R. Warmka, R., Lindell, A., and Kumar, V.: A daily global mesoscale ocean eddy dataset from satellite altimetry, Sci. Data, 2, 150 028, https://doi.org/10.1038/sdata.2015.28, 2015.

Martínez-Moreno, J., Hogg, A. M., Kiss, A. E., Constantinou, N. C., and Morrison, A. K.: Kinetic energy of eddy-like features from sea surface altimetry, J. Adv. Model. Earth Syst., 11, 3090–3105, https://doi.org/https://doi.org/10.1029/2019MS001769, 2019.

Pegliasco, C., Delepoulle, A., Mason, E., Morrow, R., Faugère, Y., and Dibarboure, G.: META3.1exp: a new global mesoscale eddy trajectory atlas derived from altimetry, Earth Syst. Sci. Data, 14, 1087–1107, https://doi.org/10.5194/essd-14-1087-2022, 2022.

Tian, F., Wu, D., Yuan, L., and Chen, G.: Impacts of the efficiencies of identification and tracking algorithms on the statistical properties of global mesoscale eddies using merged altimeter data, Int. J. of Remote Sens., 41, 2835–2860, https://doi.org/10.1080/01431161.2019.1694724, 2020.

Zhang, Z., Zhang, Y., Wang, W., and Huang, R. X.: Universal structure of mesoscale eddies in the ocean, Geophys. Res. Lett., 40, 3677–3681, https://doi.org/10.1002/grl.50736, 2013.